# Usability of Clinical Information in Discharge Summary Data in the Diagnosis Procedure Combination Survey for Cancer Patients

**DOI:** 10.3390/ijerph17020521

**Published:** 2020-01-14

**Authors:** Ayako Okuyama, Takahiro Higashi

**Affiliations:** Center for Cancer Registries, Centre for Cancer Control and Information Services, National Cancer Center, Tokyo 104-0045, Japan

**Keywords:** registries, quality assurance, health care, administrative claims

## Abstract

Valid data are required to monitor and measure the quality of cancer treatment. This study aims to assess the usability of diagnosis procedure combination (DPC) survey discharge summary data. DPC survey data were analyzed by linking them to the hospital-based cancer registries (HBCR) from 231 hospitals. We focused on patients who were aged 20 years or older and diagnosed in 2013 with stomach, colorectal, liver, lung, or breast cancer. We assessed the percentage of unknown/missing values in supplementary data for patients with five common cancers and compared DPC cancer stage information to that of HBCR. In total, 279,451 discharge data sets for 180,399 patients were analyzed. The percentages of unknown data for smoking index and height/weight were 10.5% and 2.3%, respectively, and varied from 0.0% to 93.0% between hospitals. In the activity of daily living component, the rates of missing data for climbing stairs (3.6%) and bathing (2.9%) at admission were slightly higher than for other elements. Unexpectedly low concordance rate of tumor, node, and metastasis classification between DPC survey and HBCR data was observed as 80.6%, which means 20.4% of the data showed discrepancies. The usability of DPC survey discharge summary data is generally acceptable, but some variables had substantial amounts of missing values.

## 1. Introduction

Routinely collected data from electronic health records, including administrative health claims and registries, are useful resources when evaluating the real-world effectiveness and safety of medical care [1]. Such health data are a by-product of the daily operations of healthcare systems and are collected independently of specific a priori research questions [2]. These secondary data sources provide a low-cost means to answer research questions, where answers can be obtained in a relatively short time-frame and the data are more representative of routine clinical practice, and large cohorts of patients can be followed over long time periods [3]. These types of secondary data are collected for their specific purpose. For example, health insurance claims data are produced for billing, where the information is directly related to medical charge (e.g., drug used). This data is generally expected to be accurate, as there is a central system in Japan to check the adequacy of the data (All-Japan federation of national health insurance organizations). On the other hand, other information is also submitted with claims that is not directly related to the charge (e.g., patient clinical characteristics) that may be less accurate. The potential for error occurs at many points during the process of record entry and keeping [4].

The national cancer center in Japan has collected hospital-based cancer registries (HBCR) from designated cancer care hospitals (DCCHs) yearly, since 2007, in order to describe the characteristics of cancer patients treated at DCCHs [5]. A quality of care monitoring project further collected equivalent and discharge summary data from health insurance claims that were produced for the diagnosis procedure combination (DPC) survey. The DPC is a system that determines the per-diem hospital reimbursement based on the diagnosis of patients and the procedures provided to them during their hospital stay. Traditionally, the national health insurance in Japan reimburses providers based on a fee-for-service, but hospitals can instead opt to receive reimbursement based on the DPC payment systems. The DPC survey collects data equivalent to that used for fee-for-service claims, which enables an assessment of the effect of introducing DPC payment on resource use. Since DPC surveys are automatically used through a traditional fee-for-service system, they contain data on all health services. When these data sets are linked to the HBCR, it produces a patient list with data on the care they received [6]. The HBCR records basic information related to the patient’s cancer, including topography, morphology, and tumor-node-metastasis (TNM) classification. In addition to health services data, the DPC survey also collects data on the hospitalization, including route of admission, discharge destination, and outcomes, as well as detailed clinical information such as height and weight, comorbidities including the presence/absence of pregnancy, coma scales, activity of daily living (ADL) scores adapted from the Barthel index [7], and cancer TNM classifications. Contrary to health services data, which are automatically collected in most facilities, discharge summary data is manually entered into the DPC survey. The quality of these clinical data can be questionable.

Observational studies, using routinely collected data, are expected to provide insights into the clinical situation [8]. To capture the clinical situation in elderly patients, in particular, the validity of clinical information, such as ADL and physical conditions, is key for monitoring and measuring cancer treatment. Although DPC survey data have frequently been used for research purposes, previous studies reported that a substantial proportion of DPC survey discharge summary data are missing [9,10], and the accuracy of these data are seldom discussed by comparing them to other data sources for consistency. Our previous research investigated the concordance rate of TNM classification between HBCR and DPC survey discharge data using data from four hospitals, which suggested that the concordance rate was different among hospitals [11]. These studies indicated that we need a systematic evaluation for quality of the DPC survey discharge summary data. In this study, we assessed the quality of data relevant to cancer treatment in the DPC survey discharge summary data by looking at the frequency of missing data, and its consistency with other variables in the data, national statistics and concordance with HBCR for patients with five major cancers in Japan. In addition, to determine whether these data can be used to monitor the quality of health care in each hospital, we examined differences between these missing values and the concordance rate of cancer stage by patients’ characteristics and by hospital. We analyzed the data for patients diagnosed in 2013, which were the latest published data of DPC linked to HBCR for the time being contained in the official reports published in 2017.

## 2. Materials and Methods

### 2.1. Data Source

We used data from 231 hospitals that provided DPC survey discharge summary data along with the HBCR. The DPC survey data included supplementary discharge summary data, called “Form 1”, which summarized patient’s clinical conditions during each hospitalization. Among the participating hospitals, 214 (93%) were among 409 DCCHs in the study period. We analyzed the data for patients over the age of 20 years who were diagnosed in 2013 with stomach, colorectal, liver, lung, or breast cancer, and received a first course of cancer treatment at the hospital. We used hospitalization data from 1st October 2012 through to 31st December 2014 for these patients.

### 2.2. Data Analysis

First, we identified the targeted cancer cases using HBCR. We extracted form 1 data from the DPC survey data corresponding to the identified cases and examined the usability of clinical information. Form 1 (version 2013) included 19 basic information items (e.g., height and weight, smoking index, ADL at admission, main disease, and union for international cancer control (UICC) TNM classification, and Japan coma scale (JCS)), and 11 severity information items related to specific disease (e.g., modified Rankin scale). We examined the frequency of unknown/missing data and outliers for seven basic information items (height and weight, smoking index, ADL at admission, ADL at discharge, cancer stage, JCS at admission, and JCS at discharge). We also assessed the accuracy of DPC data using either internally or externally available data. First, ADL data were assessed by a logical check against JCS. The inconsistencies of clinical information were defined as follows: (1) JCS score is registered as ten points (only awake with stimulation) or more, but one of the ADL scores is registered as an independent case; or (2) ADL score of plain walking is registered as needs assistance, but ADL score if climbing stairs is registered as independent. Secondly, we analyzed the height and weight data for outliers and their average values were compared to the national average, which was obtained from the Japanese national health and nutrition survey 2013 in Japan [12].

Finally, we compared the UICC TNM classification in the DPC survey data with those in the HBCR. The unit of record for TNM in the HBCR was newly diagnosed ‘tumor’, whereas that in the DPC survey data was ‘hospitalization’. The HBCR records the TNM before the treatment (clinical TNM, or cTNM) and after resection with pathological examination (pathological TNM, pTNM), according to the UICC rules 7th version [13]. Since the DPC survey data records TNM each time a patient is discharged from the hospital, we could not pair the HBCR and DPC TNM classifications one-on-one since some patients had more than one hospitalization during the study period. Instead, we used a lenient criterion by considering a case as concordant if the TNM classification in the DPC survey data of at least one hospitalization was the same as either the clinical or pathological stage in the HBCR [11]. We excluded cases in which the TNM classifications were unknown in both the HBCR and DPC survey data for all instances of hospitalization for the patient. Since TNM classification in HBCRs uses the UICC TNM 7th version, those patients diagnosed in 2013 DPC survey discharge summary data, which were specified as being recorded according to the UICC TNM 6th version, were excluded. We also examined differences in the concordance rate by hospital.

The frequency of unknown/missing data for each variable was evaluated across different patient characteristics, clinical departments, and hospitals. The standard errors, and thus the statistical significance levels, were adjusted by clustering the patients using Huber–White estimators. All statistical analyses were conducted using Stata 14 (Stata Corporation, College Station, TX, USA).

### 2.3. Ethical Considerations

The study was performed as part of the quality measurement/practice patterns project for cancer care that was approved by the institutional review board at the national cancer center, Japan.

## 3. Results

### 3.1. Sample Characteristics

In total, 279,451 discharges for 180,399 patients were analyzed (Table 1). The data set consisted of 48,297 stomach cancer patients (26.8%), 43,310 lung cancer patients (24.0%), 38,881 colorectal cancer patients (21.6%), 36,097 breast cancer patients (20.0%), and 13,814 liver cancer patients (7.7%). About 32.7% of patients were 65 to 74 years old when diagnosed. Approximately 265,196 to 279,451 pieces of discharge data, which depended on the clinical information provided, were used for our analyses. In total, 50.5% was discharge data was obtained from patients who were discharged from the department of internal medicine, 45.7% from patients discharged from the department of surgery, and the remaining 0.2% from those patients discharged from the emergency department.

### 3.2. Pecentage of Unknown/Missing Data in Clinical Information (Height/Weight, Smoking Index, Activity of Daily Living)

Table 2 shows the percentage of unknown height/weight, smoking index, and ADL data by patients’ characteristics and clinical departments. Overall, the percentage of unknown data for smoking index and height/weight were 10.5% and 2.3%, respectively. The percentage of unknown data for all ten ADL elements were 0.3% at admission and 0.6% at discharge. These unknown rates were higher in the older age groups. The percentage of unknown data for patients who were discharged from an emergency department ranged from 1.2% to 24.0% across different hospitals and was higher than that for those patients discharged from other departments. Regarding height and weight, only 1.4% was unknown for those patients discharged from a surgical department. For breast and lung cancer patients, the percentage of unknown data for the smoking index (8.3% for breast, 9.7% for lung) was slightly lower than for other cancer patients (11.2% to 12.3%).

We assessed the proportion of unknown data among different hospitals (Figure 1). In some hospitals, the percentage of unknown data concerning the smoking index and height/weight was higher, while the percentage of unknown data for the ADL element was similar among hospitals.

Figure 2 shows the percentage of unknown data for each ADL element. The proportion of unknown ADL data at admission is slightly higher than at discharge. The percentage of unknown data was different between ADL elements. The percentage of unknown data for climbing stairs and bathing at admission were 3.6% and 2.9%, respectively, while those in transfers and dressing were both less than 1.0%.

### 3.3. Validation of Data By Comparing Other Variables/Data Sources

#### 3.3.1. Consistency of ADL Data Compared to Japan Coma Scale

When compared with other items, 160 discharge data (0.06%) were found to show inconsistencies (e.g., patients needed total assistance for plain walking, but were considered able to climb stairs independently). In addition, 271 discharge data showed an inconsistency between ADL and JCS at admission or discharge (e.g., patients with over ten points for JCS (indicating they were only awake with stimulation) but had at least one independent ADL element).

#### 3.3.2. Height and Weight Data as Compared to the National Statistics

Figure 3 and Figure 4 shows the distribution of height and weight by different age groups, respectively. The mean height and weight were almost the same as the Japanese average in 2013. There are some outliers in the discharge data. For example, the height from the data of eight patient discharges was recorded as being under 100 cm.

#### 3.3.3. Tumour-Node-Metastasis Classifications Compared to the Hospital-Based Cancer Registries Data

The concordance rate between the DPC survey TNM classification and that of HBCR was 80.6% (95% CI; 80.3–80.8) (Table 3). The concordance rate of the M classification tended to be higher than of T and N classifications for each cancer. The concordance rate of all TNM classifications for lung (85.9%) and breast cancers (83.6%) were slightly higher than for colorectal (72.9%) and liver cancers (67.9%). The concordance rate of all TNM classifications for all five cancer types showed large variation from 40.8% to 100.0% by hospitals (Figure 5).

## 4. Discussion

The DPC survey data linked with the HBCR are an important resource for monitoring cancer treatment. This study illustrates that a certain proportion of unknown smoking index and height/weight exists in clinical information from the DPC survey, and registered clinical information, such as height and weight, were within an acceptable range. However, the percentage of unknown smoking index and height/weight was much higher in some hospitals than in others. Researchers need to be careful with this data when analyzing it at the facility level. Several studies have demonstrated the potential for the introduction of bias and a loss of precision because missing data was ignored, or inappropriate methods of handling missing values were used [14,15,16,17]. Our study illustrates that the proportion of unknown current clinical information from the DPC survey was different among hospitals. We must appropriately handle these unknown/missing values and report how we handled such data [18,19].

The proportion of DPC survey unknowns in the clinical data was slightly higher in older compared to younger populations. In this study, the proportion of unknowns in the DPC survey clinical data were higher in emergency departments than others, similar to results from previous studies of DPC survey data [9,10]. Such unknown/missing data are a common problem, even with the most rigorous of clinical data collection efforts [14].

The evaluation of clinical information by comparison to other variables or external data sets found little discordance, other than that observed for UICC TNM classification. Almost all ADL information was logically consistent (i.e., JCS and ADL). In addition, the height and weight data from the DPC survey were generally comparable to the Japanese average height and weight in 2013 [12], although a few outliers were found. Researchers should be aware of these outliers and handle them appropriately. Plausibility thresholds of height and weight can be difficult to define. Health information managers should reconfirm the extra ordinal height and weight values (e.g., less than or greater than twice the standard deviation of the survey mean).

The concordance rate between TNM classifications in the DPC survey and the HBCR was slightly higher than that of our previous study (77%, 95% CI: 75–79 in the previous study) [11]. However, we found that the concordance level varied widely across hospitals. The UICC TNM classifications are different from the regular Japanese staging systems. In HBCR, certified cancer registrars are trained to use the UICC coding system and enter TNM classifications into the records on their own by referring to clinical information, but there is no such qualification requirement to enter the same data into the DPC survey. Those hospitals with high concordance rates will be using cancer registrars to also enter data into the DPC survey. In such hospitals, the quality of data will be high. Considering these points, our results, using the data from more than 200 hospitals, indicate the need for better quality control of TNM staging information in DPC survey data. Hospital managers and health information managers should carefully record the clinical information in the DPC survey, as researchers and policy makers expect to use its survey for assessing health care quality [9,10,20,21,22].

The present study has several limitations. Firstly, we could not firmly conclude the accuracy of the clinical data in the DPC survey because we did not compare this with gold standard data (e.g., medical records). Our validation process revealed that there is no apparent problem in ADL and height/weight data compared to internal data (e.g., logically consistent between ADL and JCS) or the national health and nutrition survey. However, we found a tangible discrepancy in TNM classification between HBCR and the DPC survey. There is a chance that further inaccurate data have not yet been identified. Secondly, we analyzed the data only for patients diagnosed with cancer in 2013. The quality of clinical information in the DPC survey data may have improved since then. Finally, we analyzed the data from 231 hospitals participating in the research project and who provided clinical information to the DPC survey, to monitor quality indicators. These hospitals could have had a greater motivation towards improving the quality of health care and greater quality control of their data. Despite these limitations, this study illustrated the significance and critical points of using the clinical data in the DPC survey for monitoring quality in health care.

## 5. Conclusions

This study illustrates that unknown/missing data, outliers, and logical inconsistencies exist among clinical information in the DPC survey. When researchers and policy makers apply this data for measuring the quality of health care, they should attempt to appropriately confront this problem, and report their methods of handling the data, including how data cleaning was conducted [18,19]. Researchers should evaluate the impact of data inconsistencies and should be contrasted to typical minimally relevant effect sizes in clinical research. Such information is important as it allows readers to evaluate the strengths and weaknesses of a study. Hospital managers and health information managers should carefully record clinical information in the DPC survey, as well as conduct a logical check at the time of recording. For example, electronic case report forms (eCRFs) or other tools with automatic plausibility checks may be worthwhile, despite their costs.

## Figures and Tables

**Figure 1 ijerph-17-00521-f001:**
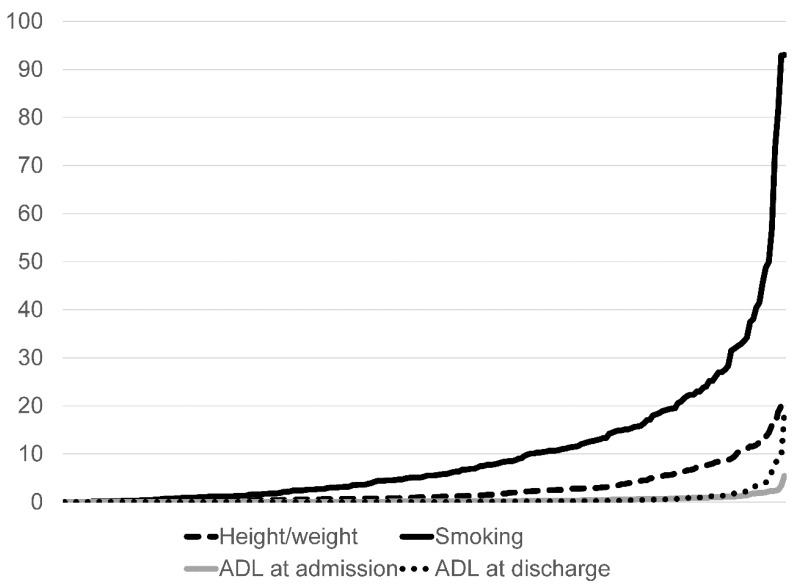
Percentage of unknown data by hospital (N = 231). ADL: Activity of daily living.

**Figure 2 ijerph-17-00521-f002:**
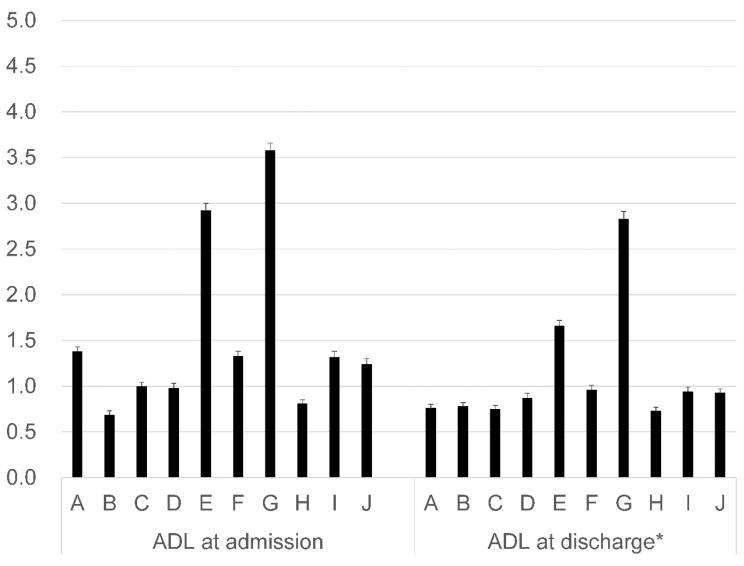
Percentage of unknown data for each ADL element (%). A: Feeding; B: Transfers; C: Grooming; D: Toileting; E: Bathing; F: Locomotion; G: Climbing stairs; H: Dressing; I: Defecation management; and J: Urination management. * Dead cases and those cases that were consulted with an obstetrician were excluded. This includes all discharge cases, even those for whom the ADL elements were unknown.

**Figure 3 ijerph-17-00521-f003:**
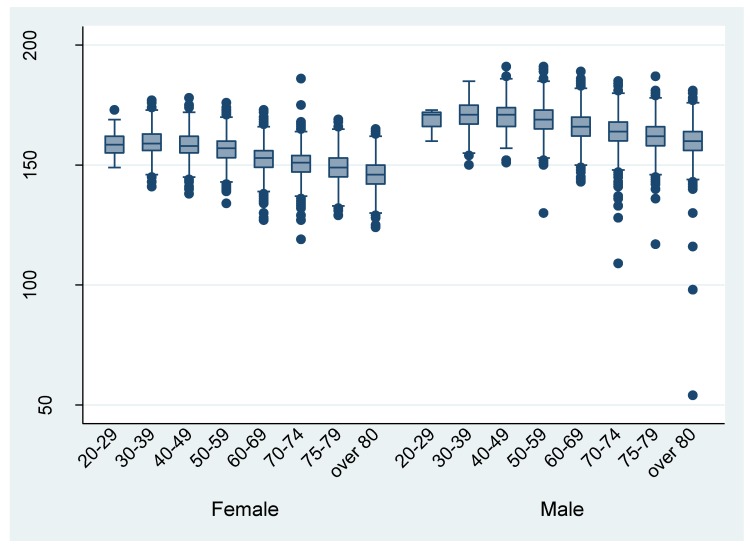
Height by different age groups (per discharge).

**Figure 4 ijerph-17-00521-f004:**
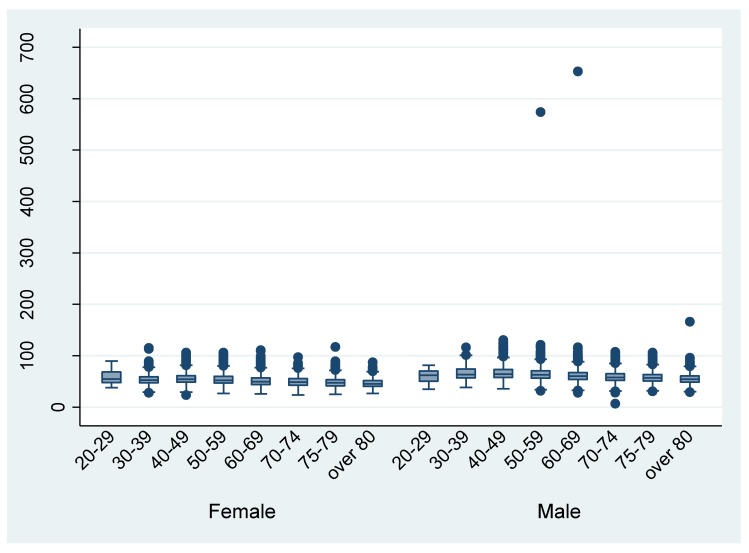
Weight by different age groups (per discharge).

**Figure 5 ijerph-17-00521-f005:**
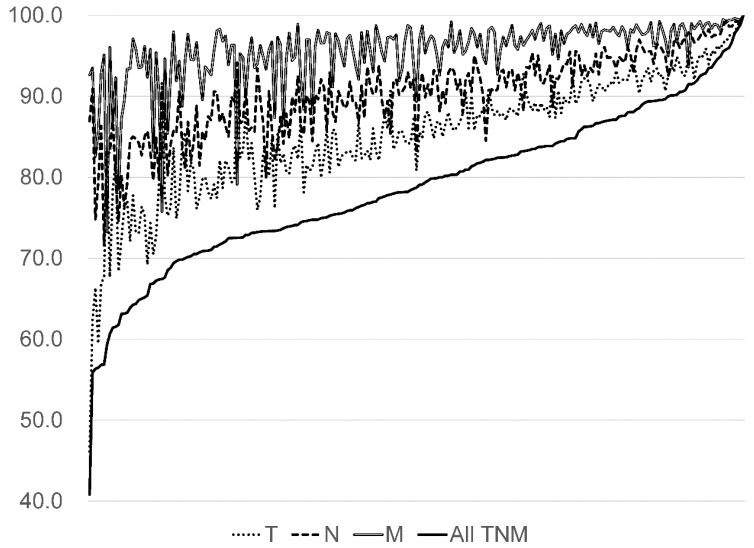
The concordance rates for all five cancers by hospital (%, N = 231).

**Table 1 ijerph-17-00521-t001:** Sample characteristics in 231 hospitals.

***Patients’ Characteristics (Per Patient)***	***n***	**(%)**
Men	95,872	53.1
Age		
20–40 years	3514	2.0
40–64 years	56,928	31.6
65–74 years	59,021	32.7
75–84 years	49,408	27.4
over 85 years	11,528	6.4
Cancer		
Stomach	48,297	26.8
Colorectal	38,881	21.6
Liver	13,814	7.7
Lung	43,310	24.0
Breast	36,097	20.0
Clinical stage		
0	11,335	6.3
I	76,386	42.3
II	29,639	16.4
III	21,762	12.1
IV	29,324	16.3
Unknown	11,953	6.6
***Clinical department (per discharge)*** *****		
Internal medicine	141,118	50.5
Surgery	127,682	45.7
Emergency department	480	0.2
Others	9826	3.5

* Number of discharge data for height and weight.

**Table 2 ijerph-17-00521-t002:** Percentage of unknown/missing data (per discharge).

	Proportion of Unknown Data, % (95% CI)			
	Height/Weight	Smoking Index	ADL at Admission *	ADL at Discharge *^,^**
	(*n* = 279,451)	(*n* = 279,451)	(*n* = 279,343)	(*n* = 265,196)
**Total**	2.3 (2.2–2.4)	10.5(10.3–10.7)	0.3(0.3–0.4)	0.6(0.6–0.6)
***Sex***				
Men	2.4(2.3–2.5)	11.9(11.6–12.1)	0.4(0.3–0.4)	0.6(0.6–0.7)
Women	2.2(2.1–2.3)	8.5(8.3–8.7)	0.3(0.3–0.3)	0.5(0.5–0.6)
***Patients’ age***				
20–40 years	0.9(0.7–1.2)	8.5(7.3–9.7)	0.2(0.1–0.4)	1.0(0.7–1.3)
40–64 years	1.4(1.3–1.5)	9.3(8.9–9.6)	0.2(0.2–0.3)	0.5(0.5–0.6)
65–74 years	1.8(1.8–1.9)	10.6(10.3–10.9)	0.3(0.3–0.3)	0.6(0.5–0.6)
75–84 years	3.1(2.9–3.2)	11.7(11.3–12.0)	0.5(0.4–0.5)	0.7(0.6–0.7)
Over 85 years	7.6(7.1–8.2)	12.1(11.4–12.8)	0.8(0.6–1.0)	0.7(0.5–0.9)
***Cancer***				
Stomach	2.3(2.2–2.4)	11.5(11.1–11.9)	0.3(0.3–0.4)	0.6(0.6–0.7)
Colorectal	2.7(2.5–2.8)	11.2(10.8–11.6)	0.4(0.4–0.5)	0.5(0.4–0.6)
Liver	2.7(2.5–3.0)	12.3(11.7–13.0)	0.4(0.3–0.5)	0.7(0.6–0.9)
Lung	2.4(2.3–2.5)	9.7(9.4–10.0)	0.3(0.3–0.4)	0.7(0.6–0.7)
Breast	1.2(1.1–1.3)	8.3(7.9–8.8)	0.2(0.1–0.2)	0.4(0.3–0.4)
***Clinical stage***				
0	1.8(1.6–2.1)	9.0(8.4–9.7)	0.2(0.1–0.3)	0.3(0.2–0.5)
I	1.7(1.6–1.8)	10.4(10.2–10.7)	0.2(0.2–0.3)	0.6(0.5–0.6)
II	1.9(1.8–2.0)	9.8(9.3–10.2)	0.3(0.3–0.4)	0.6(0.5–0.6)
III	2.3(2.2–2.5)	10.8(10.3–11.4)	0.4(0.4–0.5)	0.6(0.5–0.6)
IV	3.1(3.0–3.3)	10.7(10.3–11.2)	0.4(0.4–0.5)	0.7(0.7–0.8)
Unknown	4.1(3.7–4.5)	12.3(11.5–13.1)	0.5(0.4–0.6)	0.5(0.4–0.7)
***Clinical department***				
Internal medicine	3.0(2.9–3.1)	10.6(10.3–10.8)	0.4(0.4–0.4)	0.6(0.6–0.7)
Surgery	1.4(1.4–1.5)	10.1(9.9–10.3)	0.3(0.2–0.3)	0.5(0.5–0.6)
Emergency department	15.8(12.8–19.4)	24.0(20.4–28.0)	7.7(5.7–10.4)	1.2(0.5–3.3)
Others	2.5(2.2–2.8)	13.5(12.4–14.8)	0.3(0.2–0.5)	0.7(0.6–1.0)

ADL; Activity of daily living consists of ten elements (e.g., feeding, transfers, and grooming). * Percentage of all ten ADL elements were unknown. ** Dead cases and cases that were consulted with an obstetrician were excluded.

**Table 3 ijerph-17-00521-t003:** The concordance rates of Tumor Node Metastasis classification.

	N	Concordance Rate (%)	(95% Confidence Interval)
*All five cancers*			
T	103,214	86.8	(86.6–87.0)
N	103,869	91.3	(91.1–91.5)
M	104,088	95.8	(95.7–96.0)
All TNM	103,142	80.6	(80.3–80.8)
*Stomach*			
T	29,189	88.8	(88.5–89.2)
N	29,464	90.1	(89.7–90.4)
M	29,513	95.2	(95.0–95.5)
All TNM	29,175	82.0	(81.6–82.4)
*Colorecta*			
T	19,017	82.1	(81.6–82.7)
N	19,179	86.8	(86.3–87.3)
M	19,218	94.5	(94.1–94.8)
All TNM	19,000	72.9	(72.2–73.5)
*Liver*			
T	7697	70.9	(69.9–71.92)
N	7731	93.6	(93.1–94.2)
M	7786	94.8	(94.3–95.3)
All TNM	7681	67.9	(66.9–69.0)
*Lung*			
T	26,184	91.1	(90.7–91.4)
N	26,358	93.4	(93.1–93.7)
M	26,430	97.0	(96.8–97.2)
All TNM	26,163	85.9	(85.5–86.3)
*Breast*			
T	21,127	88.6	(88.2–89.0)
N	21,137	93.6	(93.3–94.0)
M	21,141	96.8	(96.6–97.1)
All TNM	21,123	83.6	(83.1–84.1)

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
