# Peer review of "Usability of Clinical Information in Discharge Summary Data in the Diagnosis Procedure Combination Survey for Cancer Patients"

_ijerph, 2020, doi:10.3390/ijerph17020521_

Round 1
Reviewer 1 Report
Overall I found this manuscript interesting as a data scientist analyzing clinical data frequently. Data quality and accessibility (or rather lack of it) are big hurdles in many analyses.
Abstract:
After giving some details of results, usability is deemed acceptable but missingness is states´d as problematic. The concordance rate of 80.6% (translated into some 20% error rate) is of much greater concern and warrants some emphasis here.
Introduction:
Authors state that the value of routine data is obtainability in short time frame. Yet this analysis of data from 2014 took almost 5 years. Reflecting reasons for this would be of interest.
Data collected for billing are termed to be accurate, yet I see substantial risk of bias due to commercial interests.
Results:
Material prepared for reviewers should be easily readable, tables should not be broken and broader than pages, Figures should be of sufficient quality.
Fig. 1: What is the solid line going up to 90%? Missingness of smoking? I assume hospitals are ordered by one of the 4 data? If indeed range from 0 to 100 is needed, a log scaling of y-axis would help to distinguish in the low range.
Fig.3: show only range 100-40% or again use log-scaled y-axis.
p.7 line 176: 160 data with inconsistencies is ?%
Quantitative measures height/weight: tables gives min/max, but what is the number/% of extreme values/outliers? Boxplots could visualize effects. How are plausibility thresholds defined?
Conclusions:
The evaluate the impact of data inconsistencies / discordance, they should be contrasted to typical or minimally relevant effect sizes in clinical research.
Data entering should be discussed as well, eCRFs or other tools with automatic plausibility checks may be worthwhile despite their costs.
Author Response
Thank you for your critical comments. We have revised the manuscript in response to your comments.
|
Abstract After giving some details of results, usability is deemed acceptable but missingness is states'd as problematic. The concordance rate of 80.6% (translated into some 20% error rate) is of much greater concern and warrants some emphasis here. |
We revised as follows: 'Unexpectedly low concordane rate of Tumour, Node and Metastasis classification between DPC survey and HBCR data was observed as 80.6%, which means 20.4% of the data showed discrepancies.'
|
|
Introduction: Authors state that the value of routine data is obtainability in short time frame. Yet this analysis of data from 2014 took almost 5 years. Reflecting reasons for this would be of interest.
Data collected for billing are termed to be accurate, yet I see substantial risk of bias due to commercial interests.
|
We added following sentences in introduction: 'We analysed the data for patients diagnosed in 2013, which were the latest published data of DPC linked to HBCR for the time being contained in the official reports published in 2017.'
Thank you for your comments. We added explanation as follows: 'This data is generally expected to be accurate, as there is a central system in Japan to check the adequacy of the data (All-Japan Federation of National Health Insurance Organizations). On the other hand, other information is also submitted with claims that is not directly related to the charge (e.g. patient clinical characteristics) that may be less accurate.' |
|
Results: Material prepared for reviewers should be easily readable, tables should not be broken and broader than pages, Figures should be of sufficient quality. |
We corrected tables and figures. |
| Fig. 1: What is the solid line going up to 90%? Missingness of smoking? I assume hospitals are ordered by one of the 4 data? If indeed range from 0 to 100 is needed, a long scaling of y-axis would help to distinguish in the low range. | The solid line is missing of smoking information. The hospitals are orderd by missing rate of smoking information. However, this is very complicated as you mentioned. So we changed Fig1 to show simple and clearly. |
| Fig. 3: show only range 100-40% or again use log-scaled y-axis. | We changed range 100-40%. |
| p.7 line 176: 160 data with inconsistencies is ?% | We added 0.06% of inconsistencies. (line 179) |
| Quantitative measures height/weight: tables gives min/max, but what is the number/% of extreme values/outliers? Boxplots could visualize effects. How are plausibility thresholds defined? |
We changed table 3 to boxplots (figure 3 and 4) to show outliers. Plausibility thresholds is difficult to define, however, if he or she has greater or less than twice the standard deviation of the survey mean, these data should be check their medical records. We added these in discussion. 'Plausibility thresholds of height and weight can be difficult to define. Health information managers should reconfirm the extra ordinal height and weight values (e.g., less than or greater than twice the standard deviation of the survey mean).' |
|
Conclusions: The evaluate the impact of data inconsistencies / discordance, they should be contrasted to typical or minimally relevant effect sizes in clinical research. Data entering should be discussed as well, eCRFs or other tools with automatic plausibility checks may be worthwhile despite their costs. |
Thank you for your comment. We added your comments in conclusions. |
Reviewer 2 Report
Okuyama et al. performed a study on the usability of the Diagnosis Procedure Combination (DPC) survey to monitor and measure the quality of cancer care in Japan. The DPC survey contains data on all delivered health services to patients, as well as some patient and clinical information. The authors assessed the proportions of missing/unknown values in the data from the DPC survey for height and weight, smoking index, activities of daily living (ADL), and the stage of cancer in patients with the five most common malignant diseases (i.e. stomach, lung colorectal, breast, and liver cancer). Besides, they compared DPC cancer stage information to that of hospital-based cancer registries (HBCR), of which the latter is considered the gold standard for cancer surveillance activities. Lastly, the authors assessed whether DPC survey data can be utilized to monitor and measure the quality of cancer care at the hospital level.
In the overall series, the proportion of unknown values was within an acceptable range. The proportion of unknown values tended to be higher in emergency departments than in other departments and in older cancer patients compared to younger patients. More importantly, the percentage of unknown values, particularly for the smoking index and height and weight, varied considerably across hospitals. Lastly, the concordance rate between the stage of cancer in the DPC survey and the HBCR was comparatively high. That rate, however, also varied markedly across hospitals. Collectively, DPC survey data should be used with caution to monitor and measure the quality of cancer care in Japan at the hospital level, especially in hospitals with a comparatively high number of unknown values.
The manuscript is written in a readable and accessible manner. The abstract appropriately describes the results within in the germane of the data found. The statistical methods employed are sound. The limitations of the study are discussed adequately. I have no suggestions to improve the manuscript.
Author Response
Thank you very much for your review.